# EZ-CLIP: Efficient zero-shot video action recognition

## ABSTRACT

Recent advancements in large-scale pre-training of visual-language models on paired image-text data have demonstrated impressive generalization capabilities for zero-shot tasks. Building on this success, efforts have been made to adapt these image-based visual-language models, such as CLIP, for videos extending their zero-shot capabilities to the video domain. While these adaptations have shown promising results, they come at a significant computational cost and struggle with effectively modeling the crucial temporal aspects inherent to the video domain. In this study, we present EZ-CLIP, a simple and efficient adaptation of CLIP that addresses these challenges. EZ-CLIP leverages temporal visual prompting for seamless temporal adaptation, requiring no fundamental alterations to the core CLIP architecture while preserving its remarkable generalization abilities. Moreover, we introduce a novel learning objective that guides the temporal visual prompts to focus on capturing motion, thereby enhancing its learning capabilities from video data. We conducted extensive experiments on five different benchmark datasets, thoroughly evaluating EZ-CLIP for zero-shot learning and base-to-novel video action recognition, and also demonstrating its potential for few-shot generalization. Impressively, with a mere 5.2 million learnable parameters (as opposed to the 71.1 million in the prior best model), EZ-CLIP can be efficiently trained on a single GPU, outperforming existing approaches in several evaluations.

## 1 INTRODUCTION

Large-scale pre-training of visual-language (VL) models using image-text pairs has been proven highly effective across various downstream tasks, exhibiting remarkable generalization capabilities, especially in zero-shot settings Radford et al. (2021). These models, exemplified by CLIP Radford et al. (2021) and ALIGN Jia et al. (2021), leverage extensive internet-sourced data to offer robust representations with versatile transfer and generalization capabilities.

However, extending such a pre-training strategy to videos poses significant challenges. Unlike image-text pairs, aligned video-text data is scarce, and curating it is challenging Jia et al. (2021); Xu et al. (2021). Furthermore, videos inherently possess complexity and entail substantial computational costs, while appearance cues can be captured efficiently through image-text pairs with a much lower compute budget. Therefore, adaptation of these image-language models for video-based tasks, while retaining their generic multimodal learned representations, is a promising research direction.

Motivated by this, recent works in video domain have adapted image-based visual-language models such as CLIP Radford et al. (2021) for video representation learning. The main idea is to introduce additional learnable components for spatio-temporal modeling such as self-attention layers for cross-frame communication Ju et al. (2022), textual or visual prompts Wang et al. (2021), or dedicated video encoder modules Ni et al. (2022). In a more recent effort Rasheed et al. (2023), the authors propose to fine-tune the complete CLIP model specifically for video tasks. These existing approaches have shown promising performance, however, they are computationally expensive and struggle in temporal modeling. The temporal adaptation techniques require a lot of parameters with fine-tuning of the whole model which is not desirable for preserving the generalization capability.

In light of these limitations, we propose EZ-CLIP, an efficient adaptation of visual-language models for zero-shot action recognition. EZ-CLIP introduces temporal visual prompting coupled with a simple yet effective motion constraint to address the modeling of temporal aspects in videos. The

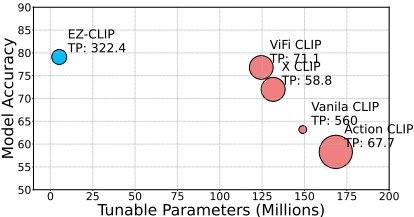 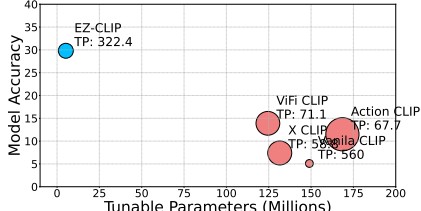

Figure 1: **Effectiveness of EZ-CLIP:** (Left) Performance comparison on UCF-101 dataset for zero-shot evaluation. The proposed method outperforms existing works while requiring fewer tunable parameters and GFLOPS with higher throughput (TP). Bubble size indicates GFLOPS during inference. (Right) Performance comparison on Something-something-v2 (SSv2) for base-to-novel evaluation. EZ-CLIP achieves significant improvement with fewer tunable parameters over existing works on this challenging dataset where motion plays a critical role.

proposed approach is computationally efficient and can be trained on a single GPU with minimal learnable parameters. EZ-CLIP does not update spatial features learned during pretraining, which preserves the generalization capability. We provide extensive evaluations across five different action recognition benchmark datasets (Kinetics-400, Kinetics-600, UCF-101, HMDB-51, and Something-something-v2) for zero-shot, base-to-novel, and few-shot settings, showcasing our model's robust generalization capability, particularly in scenarios where motion plays a crucial role (Figure 1). The main contributions of our work are as follows:

- We propose efficient adaptation of image-based visual-language models for zero-shot video action recognition, achieved with minimal learnable parameters.
- We introduce temporal visual prompting to model temporal aspect in videos. It effectively learns temporal dependencies across video frame with minimal learnable parameters.
- We propose a simple yet effective motion loss which helps in learning temporal behavior.
- Extensive evaluation across multiple action recognition datasets demonstrate robust generalization capabilities for zero-shot as well as few-shot learning. EZ-CLIP outperforms previous best methods in most evaluations with merely 5.2 million learnable parameters.

## 2 RELATED WORK

**Video action recognition** Effective models for video understanding must incorporate both spatial and motion cues. Emerging vision-transformer networks have effectively captured long-range spatio-temporal relationships, consistently demonstrating improvements over 3D CNNs Carreira & Zisserman (2017); Wang et al. (2017); Feichtenhofer et al. (2019); Christoph & Pinz (2016). In contrast to the conventional emphasis on isolated uni-modal solutions, approaches like ActionCLIP Wang et al. (2021), XCLIP Ni et al. (2022), and the study conducted by Ju et al. Ju et al. (2022) have adopted a multi-modal strategy, harnessing the potential of image-based visual-language models and applying it to zero-shot video understanding tasks.

**Vision language models** The effectiveness of learning multi-modal representations through large-scale image-text pretraining has been well-established, with demonstrated efficacy across a broad spectrum of both uni-modal and multi-modal applications Chen et al. (2020); Kamath et al. (2021); Li et al. (2019; 2020). Vision and Language (VL) models like CLIP Radford et al. (2021) and ALIGN Jia et al. (2021) pioneered the field, relying on extensive training on image-caption pairs utilizing contrastive self-supervised objectives. These models exhibit impressive transfer capabilities in downstream vision applications, encompassing few-shot and zero-shot recognition, detection, and image segmentation. However, extending these models to the domain of videos has posed challenges due to the absence of video-specific temporal cues in image-level pretraining. Recent efforts Ju et al. (2022); Ni et al. (2022); Wang et al. (2021) have sought to adapt CLIP for video applications by incorporating additional learnable components, including self-attention layers, textual or vision prompts, or dedicated visual decoders, demonstrating improvements in video-related tasks.

**Finetuning via efficient prompting** An emerging technique, prompting, offers an efficient means of applying a pre-trained model to downstream tasks without requiring the retraining of the model's existing parameters. It involves the introduction of a small number of additional learnable tokens at the model's inputs, with the primary goal of preserving the model's generalization capabilities

while enhancing its capability for downstream tasks. Originally derived from the domain of Natural Language Processing (NLP), prompting has gained widespread adoption across various vision and Vision-Language (V-L) models. CoOp Zhou et al. (2022b) and CoCoOp Zhou et al. (2022a) advocate the use of continuous learnable text prompts to facilitate the transfer of CLIP into image recognition tasks. Bahng et al. Bahng et al. (2022) introduce visual prompts as a means to probe CLIP within its visual branch. MaPLe Khattak et al. (2023) presents multi-modal prompting as an effective way to adapt CLIP while keeping the original model parameters fixed. Prompting is generally regarded as an efficient approach, demanding fewer computational resources and less training time in comparison to conventional full fine-tuning.

In the domain of video tasks, Ju et al. Ju et al. (2022) adapt CLIP through the incorporation of text prompts and transformer layers for temporal modeling. However, it is important to note that this temporal modeling, while beneficial in certain contexts, can impact CLIP's generalization ability and may pose challenges in achieving satisfactory performance in zero-shot settings. Rasheed et al. Rasheed et al. (2023) present a ViFi CLIP model that uses a fully-supervised approach and overcomes the constraints of CLIP's generalization and performance in zero-shot settings.

## 3 METHODOLOGY

We first briefly describe image-based visual language models (Sec. 3.1), then we introduce EZ-CLIP (Sec. 3.2) and describe temporal prompting (Sec. 3.2.1) followed by motion loss (Sec. 3.3) to show how we perform efficient adaptation of image-based visual-language model for video domain.

### 3.1 BACKGROUND

An image-based visual-language model consists of an image encoder $E_{image}$ and a text encoder $E_{text}$. These two encoders are trained jointly on large-scale image-text pairs by maximizing the similarity between image ($e_i$) and text ($y_i$) encodings from positive pairs. One way to achieve this is using a contrastive objective Radford et al. (2021),

$$\mathcal{L}_c(e, y) = -\sum_{i=1} \log \frac{\exp(s(e_i, y_i)/\tau)}{\sum_{j=1}^{N} \exp(s(e_i, y_j)/\tau)} \qquad (1)$$

where $s(e_i, y_i)$ is cosine similarity between image encoding $e_i$ and corresponding class encodding $y_i$, $N$ is the total number of classes and temperature parameter ($\tau$) applied to the cosine similarity between $e_i$ and $y_i$. This simple training strategy with large-scale datasets has shown remarkable zero-shot capability within these models. Once trained, these models can be used for downstream tasks such as image classification, where the target class is represented in a textual format (also termed as prompt) with the help of handcrafted templates such as '*this is a photo of [class name]*'. A simple matching between visual encodings and textual prompt of classes is then used to classify any testing sample. The ability to represent a class in form of a prompt enables zero-shot capability.

**Prompt learning and adaptation** While handcrafted prompts have showcased significant success, they suffer from sensitivity to specific templates Bragg et al. (2021). To address this limitation, the concept of prompt learning has emerged Zhou et al. (2022b). Prompt learning involves automatically learning parts of these prompts as trainable parameters, enhancing the model's robustness and generalization capabilities. In this context, consider the expression $\mathcal{Y} = \mathcal{M}_{\theta_{frozen}}([x, p])$, where $\mathcal{M}$ represents the model with frozen weights $\theta_{frozen}$, and $p$ is concatenated to the input. During training, only $p$ is tuned. This method, as highlighted in Jia et al. (2022), enables model adaptation without altering its core architecture, emphasizing its adaptability. Successful prompt learning strategies across various data types, including visual information Zhou et al. (2022b); Ju et al. (2022); Yao et al. (2021); Ge et al. (2022); Zhu et al. (2023), involve embedding learnable parameters within the model for efficient adaptation. In our approach, we introduce temporal visual prompting to capture video-specific temporal nuances, significantly boosting efficiency without compromising computational efficacy.

### 3.2 EZ-CLIP

Our goal is to efficiently adapt image-based visual-language models pre-trained on large-scale image-text pairs for video domain while preserving their zero-shot generalization capability. Since these encoders are pre-trained on image-text pair, they do not have any understanding for motion aspect in videos. To overcome this challenge, existing methods have developed special temporal

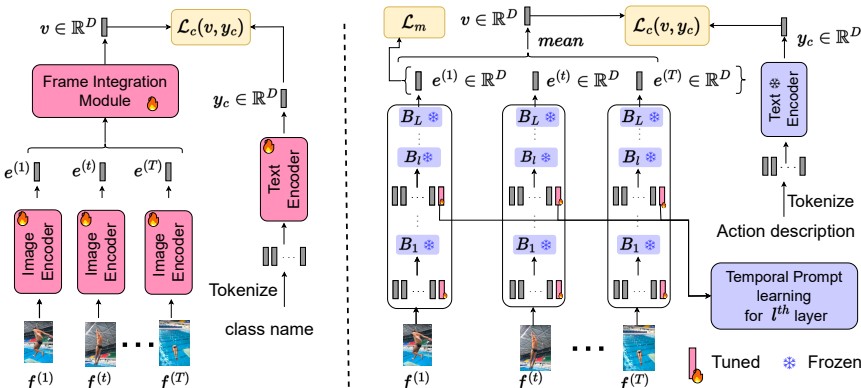

Figure 2: **Overview of the proposed method:** Proposed approach (Right) compared to adaptation-based methods (left). EZ-CLIP leverages temporal visual prompting and motion loss to efficiently learn temporal aspects, eliminating the need for the frame integration module, a bottleneck in adapting image models for video understanding. Further details on $l^{th}$ block $B_l$ (Figure 6), adapter placement (section A.3), frame processing (section 3.2.1), temporal visual prompt learning (Figure 3), and the motion loss $\mathcal{L}_{motion}$ (Section 3.3) are provided.

encoding blocks such as self-attention layers Ju et al. (2022) or dedicated video encoder modules Ni et al. (2022), which can learn the temporal relation between frames. Such approaches poses several challenges; 1) they fail to capture the temporal cues since each video frame is encoded independently, 2) they require a large amount of parameters which is an overhead on training times and compute resources, or 3) they require full fine-tuning of image backbones, which affects the generalization capability of these visual-language models.

We want to adapt visual-language to capture temporal aspect from videos while preserving the spatial learning from images with minimal modifications (without changing the already learned weights). Towards this goal, we develop EZ-CLIP, Efficient Zero-shot video action recognition model based on CLIP Radford et al. (2021), which relies on temporal visual prompting and a novel motion loss to capture motion cues from videos. We seamlessly integrate these elements into the CLIP architecture, leveraging its pre-trained capabilities. In this study, we experiment with CLIP but the proposed approach is general and should be applicable to other visual-language models.

**Problem formulation** Given a collection of video samples, we represent a video as $V = \{f^{(1)}, f^{(2)}, \ldots, f^{(T)}\} \in \mathbb{R}^{T \times H \times W \times C}$ consisting of $T$ frames each with a resolution of $H \times W$ and $C = 3$ for RGB channels. Each video is associated with a text prompt $c$ which represents the target action class. We want a model $M_v$ which can leverage visual $E_i$ and text $E_t$ encoders from image-based pre-training and provide encodings for a video $V$ and its text-pair $c$ which are similar to each other and dissimilar from prompts of other action classes.

**Overview** Given a video with a sequence of frames, we rely on the image encoder $E_i$ for encoding each frame and the text encoder $E_t$ to encode the textual prompt for action class. The visual encoder handles each frame, resulting in the generation of frame-level embeddings denoted as $\{e^{(1)}, e^{(2)}, \ldots, e^{(T)}\} \in \mathbb{R}^{T \times D}$ shown in Figure 2. Here, $e^{(t)}$ signifies the embedding of the $t^{th}$ frame within the video. The temporal visual prompts help in capturing any relations between frames to model motion aspect in videos. The spatial features are also adapted to enable effective temporal learning. These frame embeddings are then combined to create a holistic video-level representation $v \in \mathbb{R}^D$. The text encoder in image-based models also lack motion understanding therefore EZ-CLIP also adapts text encoder along with visual encoder. During training, only temporal visual prompts, spatial adapters and text adapters are trained keeping all the weights from CLIP frozen.

### 3.2.1 TEMPORAL VISUAL PROMPTING

Let's consider a video clip $V \in \mathbb{R}^{T \times H \times W \times C}$, composed of $T$ frames. Following a similar approach to CLIP image encoder Radford et al. (2021), each $t$-th frame is divided into $N$ non-overlapping patches $\{f_i^{(t)}\}_{i=1}^N$ of size $\in \mathbb{R}^{P \times P \times C}$, here, $t$ signifies the temporal index, and $N = \frac{H \times W}{P^2}$ where $P$ denotes the patch size. These patches, $\{f_i^{(t)}\}_{i=1}^N$ are then transformed into a $D-$dimensional

Figure 3: **Temporal visual prompting:** $i$) We initialize the temporal prompts $p_l \in \mathbb{R}^{T \times D}$ at the $l$-th layer of the transformer for each frame. $ii$) The association of patch embeddings with temporal prompts is accomplished using the equation 2. $iii$) The learnable temporal prompts are processed through MHA. $iv$) Processed prompts are concatenated with the $l$-th layer embeddings $[z_{l-1}^{(t)}, p_l^{(t)}]$.

patch embeddings $x_p^{(t)} \in \mathbb{R}^{N \times D}$. Then a tunable class token $x_{cls} \in \mathbb{R}^D$ is prepended to $x_p^{(t)}$ as $x_0^{(t)} = [x_{cls}; x_p^{(t)}] \in \mathbb{R}^{(N+1) \times D}$. To encode positional information, positional embeddings $E_{pos} \in \mathbb{R}^{(N+1) \times D}$ are added to $x_0^{(t)}$ as $z_0^{(t)} = x_0^{(t)} + E_{pos}$, where $z_0^{(t)}$ is the final input being fed to a sequence of transformer blocks at $t$-th frame.

To facilitate cross-frame temporal learning in a video, we introduce a novel concept called temporal visual prompting. These temporal visual prompts are strategically incorporated at the input space of each Transformer layer, denoted as $P^{Temp} \in \mathbb{R}^{L \times T \times D}$, where $L$ is number of layer in CLIP Transformer. This collection of input learnable prompts is defined as $P^{Temp} = \{p_l \in \mathbb{R}^{T \times D} | l = 0, \dots, L-1\}$. Each $t^{th}$ frame's embedding is linked to the respective temporal prompt as,

$$\tilde{p}_l^{(t)} = p_l^{(t)} + \frac{1}{N+1} \sum_{j=1}^{N+1} (z_{l-1}^{(t)})_j. \tag{2}$$

This is illustrated in Figure 3. The process then progresses into the Multi-Head Attention (MHA) mechanism, which leverages all temporal prompts to capture the global spatio-temporal dependencies within the video. This operation at the $l$-th block is expressed as:

$$\hat{p}_l = \text{MHA}(\text{LN}(\tilde{p}_l)) \tag{3}$$

where $p_l = [p_l^{(1)}, p_l^{(2)}, \dots, p_l^{(T)}]$, MHA represents a pre-trained and frozen attention block derived from CLIP, and LN denotes layer normalization. Eventually, the last learned temporal prompt is concatenated with the frame embeddings to enable subsequent processing.

**Spatial and language adaptation** To facilitate effective temporal learning, the spatial features are also adapted during training. Similarly, we also adapt the text encoder to incorporate learning of motion aspect in the textual prompts. Leveraging insights from efficient fine-tuning techniques in NLP Zhang et al. (2023); Zong et al. (2021); Zaken et al. (2021), recent advancements Zhang et al. (2023); Yang et al. (2023) have demonstrated effective strategies for fine-tuning without perturbing the original model weights. In Rasheed et al. (2023), the authors utilize prompts instead of adapters, but it requires pretraining on a video dataset which increases training overhead. In line with this, we embrace the simplicity and efficacy of the Adapter mechanism and utilize Adapters Zhang et al. (2023); Yang et al. (2023); Pan et al. (2022) to adapt both the image and text CLIP encoders. We represent action class descriptions generated by a large language model (GPT-3.5) as textual prompts. We use a template *"describe [category] as an action performed by humans"* to generate descriptions. Throughout training, all transformer layers remain frozen except for the Adapters, which are updated. Further detail of adapter placement in transformer block present at section A.3.

### 3.3 LEARNING OBJECTIVE

In the context of multimodal embedding models, our learning objective is to minimize the dissimilarity between video embeddings ($v$) and class embeddings ($y_c$) using a Contrastive loss Radford et al. (2021) (Eq. 1). The contrastive loss function is vital for training, quantifying dissimilarity between predicted and ground-truth distributions to facilitate video and class embedding association. Yet, it may overlook video's intrinsic properties. For instance, if appearance suffices for classification, it might prioritize motionless video embeddings. To tackle this, we introduce the Motion Loss, emphasizing intrinsic motion properties, enhancing motion features in video embeddings.

**Motion loss** A video is composed of a sequence of $T$ frames picked at equal interval. When there is motion in the video, generating embeddings $\{e^{(1)}, e^{(1)}, \dots, e^{(T)}\}$ for each frame results in subtle

Table 1: **Zero-shot comparison:** We assess EZ-CLIP against uni-modal approaches for zero-shot action recognition and image-based multi-modal Vision-Language (VL) models adapted for video action recognition. Performance is measured using top-1 accuracy. Models are trained on Kinetics-400 and tested on HMDB-51, UCF-101, and Kinetics-600 (disjoint classes not shared with Kinetics-400).

| Method | Input size | HMDB-51 | UCF-101 | K-600 |
|---|---|---|---|---|
| **Uni-modal zero-shot action recognition models** | | | | |
| ASR Wang & Chen (2017) | $32 \times 224 \times 224$ | 21.8 | 24.4 | – |
| ZSECOCQin et al. (2017) | $32 \times 224 \times 224$ | 22.6 | 15.1 | – |
| UR Zhu et al. (2018) | $32 \times 224 \times 224$ | 24.4 | 17.5 | – |
| E2EBrattoli et al. (2020) | $32 \times 224 \times 224$ | 32.7 | 48.0 | – |
| ER-ZSARChen & Huang (2021) | $32 \times 224 \times 224$ | 35.3 | 51.8 | – |
| **Adapting pre-trained image VL models** | | | | |
| Vanila CLIP Radford et al. (2021) | $32 \times 224 \times 224$ | 40.8 | 63.2 | 59.8 |
| ActionCLIPWang et al. (2021) | $32 \times 224 \times 224$ | 40.8 | 58.3 | 67.7 |
| XCLIP Ni et al. (2022) | $32 \times 224 \times 224$ | 44.6 | 72.0 | 65.2 |
| A5 Ju et al. (2022) | $32 \times 224 \times 224$ | 44.3 | 69.3 | 55.8 |
| ViFi CLIP Rasheed et al. (2023) | $32 \times 224 \times 224$ | 51.3 | 76.8 | **71.2** |
| EZ-CLIP(ViT-32) | $8 \times 224 \times 224$ | 50.0 | 77.5 | 67.0 |
| **EZ-CLIP(ViT-16)** | $8 \times 224 \times 224$ | **52.9** | **79.1** | 70.1 |
| **EZ-CLIP(ViT-16)** | $16 \times 224 \times 224$ | **53.3** | **80.0** | 71.1 |
| **EZ-CLIP(ViT-14)** | $8 \times 224 \times 224$ | **55.2** | **82.6** | 72.1 |

differences among these embeddings. Our goal encompasses two facets: enhancing both the diversity (variance) and the distinctiveness (central difference) among frame embeddings. This objective entails creating embeddings in a way that not only amplifies the differences between frames but also accentuates their central variations. To achieve this objective, we introduce the concept of motion loss. Let $Var$ denote the degree of diversity (variance) among the frame embeddings, and let $C$ represent the measure of central difference, then $Var$ and $C$ are defined as

$$Var = \frac{1}{T} \sum_{i=1}^{T} (e^{(i)} - e_{\text{mean}})^2, \quad \text{where} \quad e_{\text{mean}} = \frac{1}{T} \sum_{i=1}^{T} e^{(i)} \tag{4}$$

$$C = \frac{1}{T} \sum_{i=1}^{T} \frac{\partial e^{(i)}}{\partial t}, \quad \text{where} \quad \frac{\partial e^{(i)}}{\partial t} = \frac{\|e^{(i+1)} - e^{(i-1)}\|}{2}. \tag{5}$$

Our goal is to maximize the value of $\mathcal{L} = \frac{1}{dim(Var)} \sum_{i=1}^{dim(Var)} Var_i + \frac{1}{dim(C)} \sum_{i=1}^{dim(C)} C_i$ during the training process. The quantity $\mathcal{L}$ amalgamates both the desired diversity and central distinctiveness. This leads us to the formulation of motion loss:

$$\mathcal{L}_m = \frac{1}{\delta + \mathcal{L}} \quad \text{where} \quad \delta = 1. \tag{6}$$

The computed value of motion loss is inversely proportional to the sum of $\delta$ and $\mathcal{L}$. This design choice emphasizes the importance of both high diversity and substantial central differences among frame embeddings. The overall learning objective is to minimize the final loss function termed as $\mathcal{L}_{total}$ that linearly combines the traditional contrastive loss $\mathcal{L}_c$ along with $\mathcal{L}_m$ and is defined as,

$$\mathcal{L}_{total} = \lambda_1 \mathcal{L}_c(v, y_c) + \lambda_2 \mathcal{L}_m \tag{7}$$

where $\lambda_1$ and $\lambda_2$ are corresponding weights; we use equal weights for both in all our experiments.

## 4 EXPERIMENTS AND RESULTS

**Datasets:** We evaluate our proposed method on five different video action recognition benchmarks: Kinetics-400 Kay et al. (2017), Kinetics-600 Carreira et al. (2018), HMDB-51 Kuehne et al. (2011), UCF-101 Soomro et al. (2012), and Something Something V2 (SSv2) Goyal et al. (2017). Kinetics-400, Kinetics-600, HMDB-51, and UCF-101 are known to have some appearance biases where background can also be important for recognizing actions Choi et al. (2019). On the other hand, Something-something-v2 is more challenging dataset where temporal understanding is critical in recognizing the actions. Additional dataset details can be found in the Appendix A.4.

**Implementation Details:** In addition to the ViT-B/16-based CLIP model, we assess our model's generalization using CLIP ViT-32 and CLIP ViT-14 backbones. The setup employs only 8 sparsely sampled frames per video, ensuring computational efficiency. Optimization involves the AdamW

Table 2: **Base to novel generalization:** We perform comparison on four diverse datasets - Kinetics-400, HMDB-51, UCF-101, and SSv2. Models are evaluated using top-1 accuracy and HM is the harmonic mean of the base and novel classes.

| Method | Kinetics-400 | | | HMDB-51 | | | UCF-101 | | | SSv2 | | |
|---|---|---|---|---|---|---|---|---|---|---|---|---|
| | Base | Novel | HM | Base | Novel | HM | Base | Novel | HM | Base | Novel | HM |
| Vanila CLIP Radford et al. (2021) | 62.3 | 53.4 | 57.5 | 53.3 | 46.8 | 49.8 | 78.5 | 63.6 | 70.3 | 4.9 | 5.3 | 5.1 |
| ActionCLIPWang et al. (2021) | 61.0 | 46.2 | 52.6 | 69.1 | 37.3 | 48.5 | 90.1 | 58.1 | 70.7 | 13.3 | 10.1 | 11.5 |
| XCLIP Ni et al. (2022) | 74.1 | 56.4 | 64.0 | 69.4 | 45.5 | 55.0 | 89.9 | 58.9 | 71.2 | 8.5 | 6.6 | 7.4 |
| A5 Ju et al. (2022) | 69.7 | 37.6 | 48.8 | 46.2 | 16.0 | 23.8 | 90.5 | 40.4 | 55.8 | 8.3 | 5.3 | 6.4 |
| ViFi CLIP Rasheed et al. (2023) | 76.4 | 61.1 | 67.9 | 73.8 | 53.3 | 61.9 | 92.9 | 67.7 | 78.3 | 16.2 | 12.1 | 13.9 |
| EZ-CLIP(ViT-32) | 80.3 | 58.16 | 67.4 | 77.0 | 55.3 | 64.3 | 95.4 | 70.0 | 80.7 | 53.1 | 19.1 | 28.0 |
| **EZ-CLIP(ViT-16)** | **83.7** | **62.5** | **71.5** | **79.8** | **62.2** | **69.9** | **95.9** | **76.5** | **85.1** | **54.0** | **20.6** | **29.8** |
| **EZ-CLIP(ViT-14)** | **85.6** | **67.2** | **75.2** | **81.0** | **64.5** | **71.8** | **96.4** | **79.5** | **87.1** | **60.5** | **23.0** | **33.3** |

optimizer with a base learning rate of $5 \times 10^{-6}$, training for 50 epochs, and a weight decay of 0.2. The learning rate warms up for the initial 10% of epochs and then follows a cosine schedule. Training occurs on a single NVIDIA A100 80GB GPU, with a batch size of 70, and maintains an input frame resolution of $224 \times 224$ pixels. This configuration showcases the model's generalization using different backbone architectures.

## 4.1 EVALUATION AND COMPARISONS

Next, we show evaluation of EZ-CLIP for zero-shot, base-to-novel and finally its generalization capability for few-shot learning. Detail of evaluation strategy can be found in Appendix A.5.

**Zero-shot:** In this setting, the proposed model is trained on a source dataset, $D_S$ (Kinetics-400), and then directly applied to downstream cross-datasets, specifically HMDB-51, UCF-101, and Kinetics-600, for evaluation. The source dataset, $D_S$, contains samples from source classes, $Y_S = \{y_i\}_{i=0}^k$, and the evaluation is performed on the target dataset $D_T$, where $Y_S \cap Y_T = \emptyset$. We compare EZ-CLIP with both uni-modal models, and models adapting image-based multi-modal Vision-Language (VL) models (Table 1). EZ-CLIP's distinguishing feature is its consistent performance, even with training on only 8 frames per video. This efficiency is due to its ability to learn appearance and motion harmoniously. This reduces computational demands and makes our model lightweight for streamlined training and deployment. As shown in Table 1, EZ-CLIP consistently outperforms existing models, achieving substantial gains of +1.6% and +2.3% in HMDB-51 and UCF-101 respectively.

**Base-to-Novel:** To assess the generalization capabilities of EZ-CLIP towards novel classes, we evaluate it in a base-to-novel setting Rasheed et al. (2023). We begin with a dataset $D_S$ with labels $Y_S = \{y_i\}_{i=0}^k$, which is partition into two categories: the base classes $Y_B$ and the novel classes $Y_N$. This partition ensures that $Y_B \cup Y_N = Y_S$ and $Y_B \cap Y_N = \emptyset$. The model is trained on the base classes and then subjected to evaluation on both the base and novel classes. The evaluation is shown in Table 2 for four diverse datasets: K-400, HMDB-51, UCF-101, and SSv2. EZ-CLIP consistently outperforms existing models, showcasing significant improvements across all datasets. The most significant improvement is observed in SSv2 dataset, where it achieves a substantial +37.8% gain in base class accuracy and an impressive +8.6% improvement in the Novel class of SSv2.

**Generalization to few-Shot learning:** In the few-shot learning setting, we examine the model's capacity to learn under limited supervision. Given a dataset $D_S$ with labels $Y_S = \{y_i\}_{i=0}^k$, we create a general K-shot dataset where K samples are randomly selected from each category $y_i \in Y_S$ for training. We experiment with different values of K, specifically $K = 2, 4, 8,$ and 16 and use the same samples as Rasheed et al. (2023). Evaluation is performed on the validation set of $D_S$. Table 3 shows the performance of EZ-CLIP, alongside methods that adapt CLIP for video tasks. It's worth noting that while EZ-CLIP doesn't exhibit significant performance improvements due to its relatively low number of tunable parameters, it consistently maintains good performance across different shot settings outperforming existing methods in most settings.

## 4.2 ABLATIONS

In this section, we delve into the impact of temporal visual prompting, motion loss, and their combined effect on our model's performance. We use a base model with just spatial and text adapters as a baseline without temporal visual prompting and motion loss. We perform this ablation on Kinetics-400, HMDB-51, UCF-101, and Something-something-v2 datasets.

**Impact of temporal visual prompting:** We include temporal visual prompting in the base model to study its impact. We observe in Tables 4, 10 and 11 that temporal visual prompting consistently

Table 3: **Generalization to few-shot learning:** Comparison of EZ-CLIP with existing approaches on HMDB-51, UCF-101, and SSv2 Datasets across different few-shot scenarios (K = 2, 4, 6, 8 shots). Performance is evaluated using top-1 accuracy. Despite fewer tunable parameters, EZ-CLIP exhibits robust generalization abilities, with improved performance across most evaluations.

| Method | HMDB-51 | | | | UCF-101 | | | | SSv2 | | | |
|---|---|---|---|---|---|---|---|---|---|---|---|---|
| | K=2 | K=4 | K=8 | K=16 | K=2 | K=4 | K=8 | K=16 | K=2 | K=4 | K=8 | K=16 |
| Vanila CLIP Radford et al. (2021) | 41.9 | 41.9 | 41.9 | 41.9 | 63.6 | 63.6 | 63.6 | 63.6 | 2.7 | 2.7 | 2.7 | 2.7 |
| ActionCLIPWang et al. (2021) | 47.5 | 57.9 | 57.3 | 59.1 | 70.6 | 71.5 | 73.0 | 91.4 | 4.1 | 5.8 | 8.4 | 11.1 |
| XCLIP Ni et al. (2022) | 53.0 | 57.3 | 62.8 | 62.4 | 71.4 | 79.9 | 83.7 | 91.4 | 3.9 | 4.5 | 6.8 | 10.0 |
| A5 Ju et al. (2022) | 39.7 | 50.7 | 57.0 | 62.4 | 71.4 | 79.9 | 85.7 | 89.9 | 4.4 | 5.1 | 6.1 | 9.7 |
| ViFi CLIP Rasheed et al. (2023) | 57.2 | **62.7** | 64.5 | 66.8 | 80.7 | 85.1 | 90.0 | 92.7 | 6.2 | 7.4 | 8.5 | 12.4 |
| EZ-CLIP(ViT-32) | 52.4 | 56.9 | 60.4 | 63.7 | 79.7 | 83.7 | 87.5 | 89.8 | 5.8 | 6.7 | 8.0 | 12.1 |
| **EZ-CLIP(ViT-16)** | **57.3** | 61.1 | 65.4 | 67.7 | 84.7 | 88.3 | 91.3 | 92.8 | 6.8 | 8.7 | 9.6 | 13.2 |
| **EZ-CLIP(ViT-14)** | 62.4 | 65.3 | 68.4 | 70.7 | 85.2 | 89.7 | 92.4 | 93.3 | 8.9 | 10.1 | 11.1 | 15.2 |

Table 4: **Ablation study:** We evaluate the effectiveness of temporal visual prompting (TVP) and motion loss using base-to-novel setup on Kinetics-400, UCF-101, HMDB-51 and SSv2.

| Method | | Kinetics-400 | | | HMDB-51 | | | UCF-101 | | | SSv2 | | |
|---|---|---|---|---|---|---|---|---|---|---|---|---|---|
| TVP | Motion Loss | Base | Novel | HM | Base | Novel | HM | Base | Novel | HM | Base | Novel | HM |
| ✗ | ✗ | 78.3 | 56.1 | 65.3 | 74.8 | 56.3 | 64.2 | 94.4 | 74.8 | 83.4 | 30.7 | 15.4 | 20.5 |
| ✗ | ✓ | 80.8 | 59.2 | 68.3 | 76.6 | 57.3 | 65.5 | 95.0 | 75.3 | 84.0 | 34.5 | 16.3 | 22.1 |
| ✓ | ✗ | 81.2 | 60.0 | 69.0 | 78.5 | 57.8 | 66.5 | 95.6 | 76.4 | 84.9 | 45.1 | 18.5 | 26.2 |
| ✓ | ✓ | **83.7** | **62.5** | **71.5** | **79.8** | **62.2** | **69.9** | **95.9** | **76.5** | **85.1** | **54.0** | **20.6** | **29.82** |

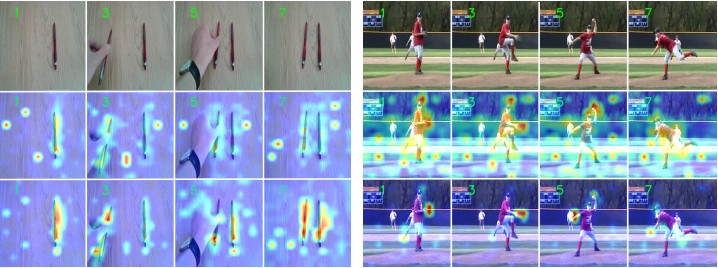

Figure 4: **Visualizing effectiveness of EZ-CLIP:** Comparison of attention maps between EZ-CLIP and a base model without temporal prompts and motion loss on examples from SSv2 (left) and UCF-101 (right) validation sets. First row: input video frames, second row: result with base model, and third row results with EZ-CLIP. Left example shows action class '*Putting something next to something*'. Base model primarily focuses on object appearance, while EZ-CLIP attends to object interactions and relative motion. Right example shows action class '*Baseball Pitch*' where base model focuses on object appearance and background features, while EZ-CLIP tracks motion regions.

improves accuracy over the base model for both base and novel classes and across all the datasets. In addition, we also observe that the improvement is more significant on SSv2 dataset in comparison with HMDB-51 and UCF-101 datasets, which signifies its temporal modeling capability.

**Impact of Motion Loss:** Adding the proposed motion loss as a constraint during base model training consistently enhances performance, albeit to a lesser extent than observed with temporal prompting (Tables 4, 10, 11). This improvement is attributed to the model's limited ability to learn temporal aspects, given the absence of temporal prompts. More detailed insights into the impact of Motion Loss on video frame embeddings are provided in Section (A.9).

**Joint training:** Finally, when we train a model jointly with both temporal visual prompts and motions loss, we observe significant improvement across all datasets; the improvement is significantly better in case of SSv2 dataset as compared with HMDB-51 and UCF-101. This further strengthens our claim that temporal prompting and motion loss can help in better temporal modeling in videos.

## 4.3 DISCUSSION AND ANALYSIS

**UCF-101 and SSv2 as extremes**: UCF-101 and SSv2 represent the opposite ends of the spectrum in action recognition datasets. UCF-101 achieves high performance without explicit motion learning, leveraging pretrained CLIP weights that emphasize appearance. However, when temporal visual prompts and motion loss are introduced, the attention shifts towards motion, as shown in the right

Table 5: **Descriptive prompts:** Impact of LLM-generated action class descriptions as prompts.

| | HMDB-51 | | | UCF-101 | | | SSv2 | | |
|---|---|---|---|---|---|---|---|---|---|
| LLM-description | Base | Novel | HM | Base | Novel | HM | Base | Novel | HM |
| ✗ | 77.3 | 59.3 | 67.1 | 94.3 | 65.5 | 77.3 | 51.7 | 18.8 | 27.57 |
| ✓ | **79.8** | **62.2** | **69.9** | **95.9** | **76.5** | **85.1** | **54.0** | **20.6** | **29.82** |

Table 6: **Efficiency of EZ-CLIP:** Comparison of EZ-CLIP with existing methods. Throughput per view (TP) is measured using a single A100 GPU. EZ-CLIP demonstrates superior efficiency in terms of GFLOPs, throughput, and parameter count, highlighting its computational advantages.

| Method | GFLOPs ↓ | TP ↑ | Total params (M) ↓ | Tunable params (M) ↓ |
|---|---|---|---|---|
| ActionCLIPWang et al. (2021) | 563 | 67.7 | 168.5 | 168.5 |
| XCLIP Ni et al. (2022) | 287 | 58.5 | 131.5 | 131.5 |
| ViFi CLIP Rasheed et al. (2023) | 281 | 71.1 | 124.7 | 124.7 |
| EZ-CLIP | **102** | **322.4** | **87.5** | **5.2** |

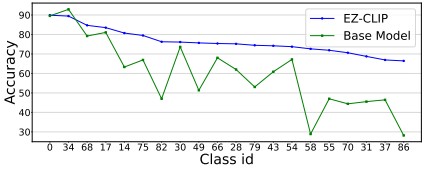 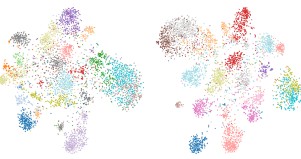

Figure 5: **Class-wise analysis:** (Left) Performance improvement with EZ-CLIP over base model (without temporal prompting and motion loss) on top-performing 20 classes. On class 0 ('*pouring something on something*'), even base model is performing well as motion is not important, whereas in 86 ('*putting something on surface*'), where motion is important, EZ-CLIP improves significantly (appearance will be same for '*picking something from surface*'). (Middle) T-sne visualization of these classes from base model, and (right) t-sne visualization of features from EZ-CLIP.

part of Figure 4. The baseline CLIP with adapters focuses on appearance and background, while EZ-CLIP tracks the motion region. In contrast, SSv2, with its high reliance on motion cues, benefits significantly in our 'base to novel' experiments. Attention visualizations in Figure 4 highlight CLIP's emphasis on background, while EZ-CLIP learns complex relative motion features.

**Template prompt vs action class description:** Tables 5,9 and 12 provides a detailed comparison of the model's performance with and without LLM-generated descriptions on two datasets: UCF-101 and HMDB-51. Our findings reveal that incorporating LLM-generated action class descriptions leads to improved text embedding generalization and enhanced model accuracy.

**Efficiency analysis:** We analyze the computational complexity of various methods (Table 6). Existing CLIP adaptation methods exhibit lower throughput due to the incorporation of video-specific learnable components alongside the vanilla CLIP model. In contrast, EZ-CLIP addresses this efficiency challenge effectively by using temporal visual prompts, which are 73K learnable inputs attached to the input space to capture temporal consistency. We have omitted the additional video-specific learning module, as shown in Figure 2. As a result, EZ-CLIP demonstrates higher efficiency in terms of throughput while maintaining comparable FLOPs compared to previous approaches.

**Class-wise performance analysis:** In Figure 5, we present a comparison of top performing 20 classes (Appendix A.7) between EZ-CLIP and a base model with 'No temporal visual prompt and no motion loss'. This comparison provides insights into our model's performance variations across different classes. Additionally, the t-SNE visualization showcases how our model with 'temporal visual prompt and motion loss' achieves better class separation in feature space.

## 5 CONCLUSION

In this work, we propose EZ-CLIP, an efficient model for zero-shot video action recognition. EZ-CLIP can effectively adapt image-based visual-language models to learn temporal aspect in videos while preserving its generalization capability. It is based on temporal visual prompting which requires very few learnable parameters making the adaptation efficient. We also propose a novel motion loss which enhances the temporal modeling capability of EZ-CLIP. With limited number of learnable parameters, EZ-CLIP can be trained on a single GPU with significantly reduced computational resources, yet consistently achieving superior performance across different evaluations.

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

# A APPENDIX

## A.1 SUPPLEMENTARY MATERIAL

We present supplementary material that enhances the understanding of our main paper through additional details and in-depth qualitative analysis. This supplementary content is structured as follows:

1. **Class Description Table**: A comprehensive table detailing the descriptions of various action classes, enhancing the clarity of our dataset.

2. **Spatial and Language Adaptation**: Delving deeper into the spatial and language adaptation techniques employed, providing an intricate understanding of our model's adaptability.

3. **Dataset Details**: Detailed information about the datasets used in our study, shedding light on their characteristics and nuances.

4. **Evaluation Protocol**: A thorough explanation of our evaluation protocols, offering insights into our methodology and ensuring transparency in our assessments.

5. **Out-of-Distribution Results**: Further analysis of out-of-distribution results, demonstrating the robustness of our model in unconventional scenarios.

6. **SSV2 Top 20 Performing Class Names**: A curated list of the top 20 performing classes in the SSV2 dataset, highlighting the model's proficiency in specific action categories.

This supplementary material enriches the main paper by providing a deeper context, detailed methodologies, and a nuanced analysis of our results.

## A.2 CLASS DESCRIPTION TABLE

In our pursuit of enhanced generalization, we utilized GPT-3.5 to generate action class names, aiming for a broader and more adaptable understanding of actions. To provide a comprehensive overview of our approach, we present a curated selection of UCF class names and their corresponding descriptions in Table 7. This table encapsulates the amalgamation of linguistic and visual comprehension, illustrating how our LLM-enhanced model interprets and defines various actions within the UCF-101 dataset.

Table 7: Description of selected UCF action classes generated by GPT-3.5.

| Action Class name | Example of GPT-3.5 Description |
| --- | --- |
| ApplyEyeMakeup | ApplyEyeMakeup is an action performed by humans to enhance their eyes and create a more dramatic look. It involves using a variety of makeup products such as eyeshadows, eyeliners, and mascaras, as well as blending and contouring techniques to create a desired effect. |
| Basketball | Basketball is a sport played by two teams of five players on a rectangular court. The objective is to shoot a ball through a hoop 18 inches in diameter and 10 feet high mounted to a backboard at each end. The game is played by bouncing the ball on a hard court surface. |
| PizzaTossing | Pizza tossing is an art form of spinning and stretching dough to create a thin and crispy pizza crust. It involves the use of hands, wrists, and arms to shape the dough into a round shape before it is placed in the oven. The goal is to achieve a consistent thickness and texture. |
| Typing | Typing is an action performed by humans using a keyboard to input text into a computer or other device. It is a skill that requires practice and repetition to become proficient. Typing can be used to create documents, send emails, type code, or enter data into databases. |

## A.3 SPATIAL AND LANGUAGE ADAPTATION

To enable effective temporal learning using temporal visual prompting, spatial features and the text encoder are adapted during training. The transformer block structure $B_l$ in Figure 6 illustrates the integration of the Adapter after the self-attention layer, adapting the CLIP architecture. Throughout training, all transformer layers remain frozen, except for the Adapters, which are updated.

For the image encoder, the computation of a transformer block is expressed as

$$[\tilde{z}_l^{(t)}, \overline{p}_l^{(t)}] = [z_{l-1}^{(t)}, \hat{p}_l^{(t)}] + \text{MHA}(\text{LN}([z_{l-1}^{(t)}, \hat{p}_l^{(t)}])) \tag{8}$$

where $[\cdot, \cdot]$ concatenates the frame embeddings and temporal prompts. For adaptation, this output is passed to an Adapter followed by a residual block,

$$\hat{z}_l^{(t)} = \text{Adapter}(\tilde{z}_l^{(t)}) \tag{9}$$

$$z_l^{(t)} = \hat{z}_l^{(t)} + \text{MLP}(\text{LN}(\hat{z}_l^{(t)})), \quad l = 0, 1 \ldots L - 1 \tag{10}$$

Consequently, we employ Adapters for fine-tuning the CLIP text encoder. The adaptation for the text encoder can be represented similarly to the image encoder,

$$\tilde{y}_c = c_{des} + \text{MHA}(\text{LN}(c_{des})) \tag{11}$$

$$\hat{y}_c = \text{Adapter}(\tilde{y}_c) \tag{12}$$

$$y_c = \hat{y}_c + \text{MLP}(\text{LN}(\hat{y}_c)) \tag{13}$$

where $c_{des}$ is the text embedding of the description of class $c$. Throughout training, all transformer layers remain frozen except for the Adapters, which are updated.

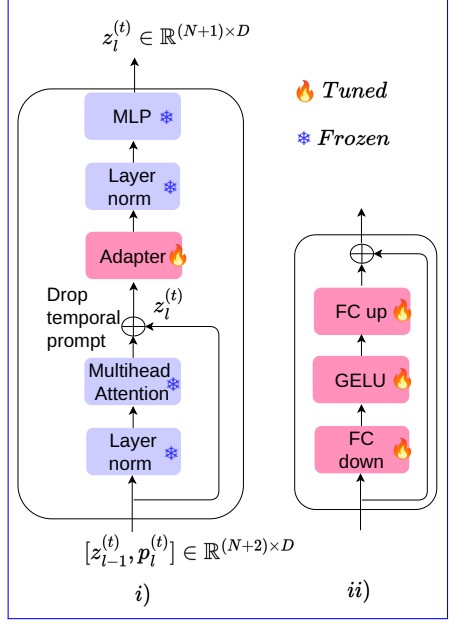

Figure 6: $l^{th}$ **layer transformer block** $B_l$: $i$) Left figure showing the $l^{th}$ transformer layer $B_l$ which takes input $[z_{l-1}^{(t)}, p_l^{(t)}] \in \mathbb{R}^{(N+2) \times D}$ after MHA. We use the adapter block to adapt the image and text model. $ii$) Right figure showing the adapter block structure, the first FC layer reduces dimensionality, while the second FC layer restores it to the original dimensions.

## A.4 DATASET DETAILS

Our analysis is conducted on five well-established action recognition datasets:

**Kinetics-400 and Kinetics-600 Carreira et al. (2018):** The Kinetics-400 dataset comprises 400 human action classes, represented by video clips sourced from various YouTube videos, each lasting around 10 seconds. It includes approximately 240,000 training videos and 20,000 validation videos. Kinetics-600 extends Kinetics-400, covering 600 action categories, with about 410,000 training video clips and 29,000 validation video clips.

**HMDB-51 Kuehne et al. (2011):** The HMDB-51 dataset contains 71,000 realistic videos collected from various sources, spanning 51 action categories. The standard split includes 3,570 training samples and 1,530 validation samples. These sets are further divided into three splits, each containing 70 training clips and 30 validation clips for each action category.

**UCF-101 Soomro et al. (2012):** UCF-101 comprises 13,000 realistic videos sourced from YouTube, encompassing 101 action categories, including five action types: human-object interaction, body-motion, human-human interaction, playing instrumental music, and sports. The standard split involves training on 9,537 videos and evaluation on 3,783 videos, distributed across three splits.

**Something Something V2 (SSv2) Goyal et al. (2017):** The SSv2 dataset is an extensive collection of video clips depicting humans performing actions with everyday objects, spanning 174 action categories. This dataset focuses on recognizing fine-grained actions, such as covering something with something or uncovering something, making it more temporally biased compared to other

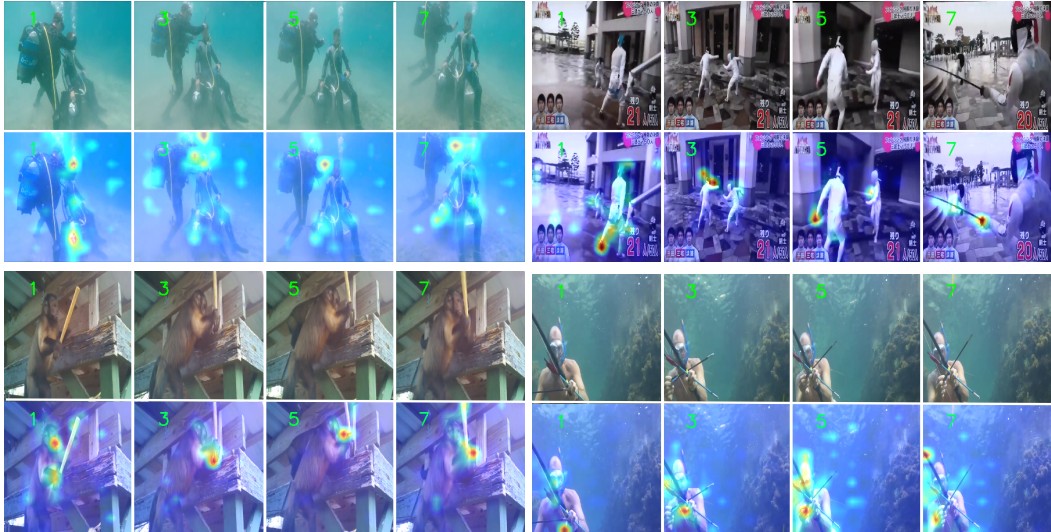

Figure 7: EZ-CLIP's attention map visualizations on extreme out-of-distribution examples from the UCF-DS dataset Schiappa et al. (2023). The model demonstrates remarkable generalizability on unconventional actions like "Underwater Haircut" (top left), "Fencing in Crowd"(top right), "Hammering by Animal"(bottom left), and "Underwater Archery"(bottom right).

datasets. The standard split consists of 168,913 training videos and 24,777 validation videos. Our reported performance metric is top-1 accuracy over the validation split.

## A.5 EVALUATION PROTOCOLS

In our analysis, we explore four distinct experimental scenarios: zero-shot, base-to-novel generalization, few-shot, and fully-supervised settings. Across these scenarios, we employ a sparse sampling approach Wang et al. (2016), capturing 8 frames consistently. Each sampled frame is resized, with the shorter side set to 256 pixels, and centered to create a 224-pixel square crop.

### A.5.1 ZERO-SHOT SETTING:

In the zero-shot setting, models trained on the Kinetics-400 dataset undergo evaluation on three different cross-datasets: HMDB-51, UCF-101, and Kinetics-600. For HMDB-51 and UCF-101, the methods are assessed across their respective three validation splits, and the top-1 average accuracy is reported. Regarding Kinetics-600, we assess the performance on 220 categories that do not overlap with Kinetics-400, reporting top-1 accuracy. In this setting, single-view inference using 8 frames is applied.

### A.5.2 BASE-TO-NOVEL SETTING:

For a comprehensive assessment of the generalization capabilities of various approaches, we adopt the base-to-novel generalization setting Rasheed et al. (2023) for video action recognition tasks. Here, a model is initially trained on a set of base (seen) classes in a few-shot manner and subsequently evaluated on a set of novel (unseen) classes. We conduct a thorough generalization analysis across four datasets: Kinetics-400, HMDB-51, UCF-101, and SSv2. The dataset employs three training splits, classifying the total categories into two equal halves. The most frequently occurring classes constitute the base classes, while the rarely occurring categories are designated as the novel classes. This setup employs 8 frames and follows a single-view inference.

### A.5.3 FEW-SHOT SETTING:

The few-shot setting involves creating a general K-shot split, with K samples used in accordance with splits from Rasheed et al. (2023). Specifically, we experiment with 2, 4, 8, and 16 shots on three datasets: HMDB-51, UCF-101, and SSv2. The models are assessed on the first validation split for HMDB-51 and UCF-101, and the full validation split, even on temporally-challenging datasets like SSv2.

### A.6 OUT-OF-DISTRIBUTION RESULTS

In our exploration of extreme out-of-distribution examples, we turned our attention to the UCF-DS dataset Schiappa et al. (2023), a collection of videos that push the boundaries of conventional contexts. These unconventional scenarios serve as litmus tests for a model's adaptability and robustness. In the peculiar world of 'Underwater Haircut,' EZ-CLIP's attention hones in on the person's head region, showcasing its ability to interpret unique actions. Venturing further, the model navigates the complex dynamics of 'Fencing in a Crowd,' deftly capturing the intricate hand motions amidst the chaos.Delving into more extraordinary territories, 'Hammering by Animal' presents a scenario where an animal performs an unexpected action. EZ-CLIP astutely focuses on the animal's hand motion, demonstrating its capacity to understand unconventional movements . Finally, in the realm of 'Underwater Archery,' the model's attention locks onto the action area with precision, highlighting its exceptional adaptability. These insightful visualizations underscore EZ-CLIP's extraordinary ability to navigate the unknown, positioning it not just as a tool for everyday scenarios but as a robust solution even in the face of the extraordinary .

### A.7 SSV2 TOP 20 PERFOMING CLASS NAMES

Table 8: Class names of top 20 performing class of SSv2

| ID | Action Description |
|----|-------------------|
| 0 | Pouring something into something |
| 34 | Plugging something into something |
| 68 | Tearing something into two pieces |
| 17 | Folding something |
| 14 | Throwing something in the air and catching it |
| 75 | Turning something upside down |
| 82 | Covering something with something |
| 30 | Something falling like a feather or paper |
| 49 | Lifting up one end of something without letting it drop down |
| 66 | Pushing something so that it falls off the table |
| 28 | Spinning something that quickly stops spinning |
| 79 | Taking one of many similar things on the table |
| 43 | Tearing something just a little bit |
| 54 | Lifting something with something on it |
| 58 | Moving something and something closer to each other |
| 55 | Moving something and something away from each other |
| 70 | Putting something similar to other things that are already on the table |
| 31 | Putting something and something on the table |
| 37 | Putting something behind something |
| 86 | Putting something on a surface |

### A.8 ADDITIONAL ABLATION EXPERIMENTS

Table 9: **Zero-shot ablation LLM:** Impact of LLM-generated action class descriptions as prompts on Zero-shot setting.

| LLM-description | HMDB-51 | UCF-101 | K-600 |
|---|---|---|---|
| ✗ | 52.0 | 77.8 | 67.2 |
| ✓ | **52.9** | **79.1** | 70.1 |

Table 10: **Zero-shot ablation** We evaluate the effectiveness of temporal visual prompting (TVP) and motion loss while using Zero shot setup train on Kinetics-400 tested on UCF-101, HMDB-51 and Kinetics-600.

| TVP | Motion loss | HMDB-51 | UCF-101 | K-600 |
|---|---|---|---|---|
| ✗ | ✗ | 49.2 | 75.5 | 63.1 |
| ✗ | ✓ | 50.0 | 75.9 | 65.5 |
| ✓ | ✗ | 51.2 | 78.1 | 67.5 |
| At first layer only | ✓ | 50.4 | 76.3 | 65.6 |
| ✓ | ✓ | **52.9** | **79.1** | **70.1** |

Table 11: **Few shot Ablation:** We evaluate the effectiveness of temporal visual prompting (TVP) and motion loss while using Few-shot setup on UCF-101, HMDB-51 and SSv2.

| Method | | HMDB-51 | | | | UCF-101 | | | | SSv2 | | | |
|---|---|---|---|---|---|---|---|---|---|---|---|---|---|
| TVP | Motion Loss | K=2 | K=4 | K=8 | K=16 | K=2 | K=4 | K=8 | K=16 | K=2 | K=4 | K=8 | K=16 |
| ✗ | ✗ | 53.7 | 57.9 | 61.7 | 63.6 | 81.2 | 85.5 | 86.6 | 89.8 | 5.7 | 6.7 | 7.9 | 10.8 |
| ✗ | ✓ | 56.0 | 58.5 | 64.7 | 65.7 | 82.1 | 84.6 | 88.5 | 90.3 | 5.9 | 7.7 | 8.7 | 12.0 |
| ✓ | ✗ | 56.7 | 60.3 | 63.5 | 66.7 | 83.3 | 86.1 | 89.9 | 91.6 | 6.6 | 8.1 | 9.0 | 12.7 |
| ✓ | ✓ | **57.3** | **61.1** | **65.4** | **67.7** | **84.7** | **88.3** | **91.3** | **92.8** | **6.8** | **8.7** | **9.6** | **13.2** |

Table 12: **Few shot Ablation LLM:** Impact of LLM-generated action class descriptions as prompts on Few-shot setting.

| LLM-description | HMDB-51 | | | | UCF-101 | | | | SSv2 | | | |
|---|---|---|---|---|---|---|---|---|---|---|---|---|
| | K=2 | K=4 | K=8 | K=16 | K=2 | K=4 | K=8 | K=16 | K=2 | K=4 | K=8 | K=16 |
| ✗ | 57.0 | 59.1 | 65.1 | 65.2 | 81.8 | 86.6 | 88.8 | 90.2 | 6.1 | 8.0 | 9.3 | 11.9 |
| ✓ | **57.3** | **61.1** | **65.4** | **67.7** | **84.7** | **88.3** | **91.3** | **92.8** | **6.8** | **8.7** | **9.6** | **13.2** |

## A.9 IMPACT OF MOTION LOSS ON VIDEO FRAME EMBEDDINGS

Motion loss plays a crucial role in enforcing frame separation within a video. Visualization of video frame embeddings in a t-Distributed Stochastic Neighbor Embedding (t-SNE) plot provides insights into the impact of motion loss.

When motion loss is not considered, the frame embeddings appear closely concentrated within the video. This concentration suggests that the model predominantly focuses on the appearance of frames while neglecting the motion aspect. However, with the incorporation of motion loss, the frame embeddings exhibit a level of variance. This variance signifies that the model does not treat all video frames identically. The model, by learning this variance within frames, gains the ability to discern motion within the video.

The t-SNE plot comparison, shown in Figure 8, visually demonstrates the differences in frame embeddings between scenarios with and without motion loss.

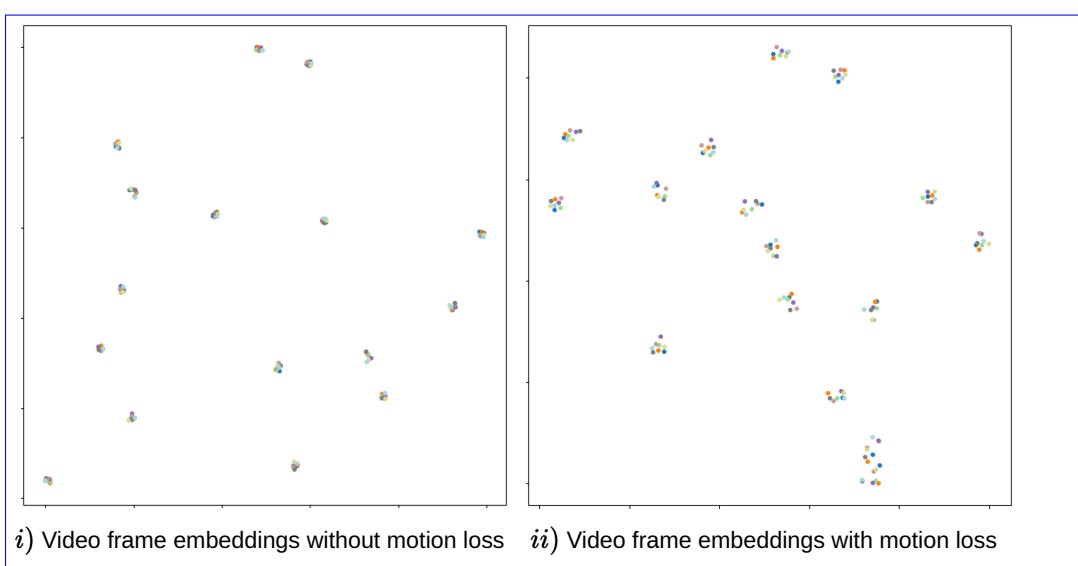

$i$) Video frame embeddings without motion loss    $ii$) Video frame embeddings with motion loss

Figure 8: Comparison of t-SNE plots for frame embeddings with and without motion loss. The plot on the left represents embeddings without motion loss, showing concentrated points, while the plot on the right depicts embeddings with motion loss, demonstrating variance and capturing motion dynamics within the video. Each of the eight different colors represents the frame embeddings of a video, and each small cluster of eight frames signifies one video.

