# OpenReview forum: "EZ-CLIP: EFFICIENT ZERO-SHOT VIDEO ACTION RECOGNITION"
_ICLR.cc/2024/Conference — Submitted to ICLR 2024_

### Official Review · Reviewer_wVRo · 2023-10-30

**Soundness:** 3 good
**Presentation:** 3 good
**Contribution:** 2 fair
**Rating:** 5
**Confidence:** 5

**Summary:**

This paper introduces EZ-CLIP, an efficient zero-shot video recognition model, which incorporates temporal visual prompting into the pretrained vision-language model CLIP to capture video motions. During model training, the newly added temporal module is the only component that is learned, making it an efficient approach. Additionally, a motion loss is introduced to enhance diversity and distinctiveness among frames, thus improving model learning. The paper's results on multiple datasets demonstrate its strong performance.

**Strengths:**

The paper effectively communicates its motivation and methodology. It highlights the key idea of utilizing the image-based visual-language model CLIP and seamlessly integrating temporal visual prompting for video action recognition.

The model's simplicity and clarity make it easily understandable. Moreover, the paper provides compelling evidence of its superiority over alternative methods through results on various benchmarks.

**Weaknesses:**

1. The authors might overstate the claim that "temporal adaptation techniques require a lot of parameters with fine-tuning of the whole model which is not desirable for preserving generalization capability." Both AIM [1] and ST-Adapter [2] employ a similar learning approach by freezing CLIP parameters and only tuning newly added adapter layers to learn temporal relations within video frames. It's crucial to maintain accuracy in comparative statements.

[1] AIM: Adapting Image Models for Efficient Video Action Recognition, https://arxiv.org/abs/2302.03024

[2] St-adapter: Parameter-efficient image-to-video transfer learning, NeurIPS, 2022

2. The used baseline models are two adapters AIM and LoRA-FA (the last paragraph of Section 3.2). The paper's baseline model utilizes the spatial adapter only, which makes it less comparable to methods like AIM that incorporate both spatial and temporal adapters. Including both spatial and temporal adapters in the baseline would provide a more accurate basis for comparing the proposed temporal visual prompting.

3. Table 5 demonstrates the substantial positive impact of the LLM-generated action class descriptions on action recognition. It suggests that the improved performance of EZ-CLIP may primarily stem from the use of these descriptions rather than the introduction of temporal visual prompting. For example, EZ-CLIP wo LLM-description only achieves 77.3% HM result on UCF-101, while ViFi CLIP which does not use LLM (I checked their paper, if I did not miss something) obtains the higher result of 78.3%. As such, the better performance of EZ-CLIP may mainly come from the utilization of LLM.


4. The proposed EZ-CLIP employs 8 frames for efficiency, while other methods use 32 frames (a 4x difference). As such, the comparison in Table 6 may not be appropriate.

5. It's worth noting that K-600 is an extension of K-400 and shares video categories. This overlap should be taken into account when doing zero-shot learning.

**Questions:**

Please see my comments in Weaknesses.

---

> ### Author Response · Authors · 2023-11-22
> **Responce to Reviewer wVRo[1/3]**
>
> We thank the reviewer for their valuable time, helpful suggestions, and questions. We are grateful to provide our responses as follows:
>
> **W3.1: Clarification on overstatement.**
>
> **Answer:**
> Thank you for bringing this to our attention. We appreciate your insightful observation and agree with the reviewer. It's important to note that our statement regarding the undesirability of fine-tuning the whole model for preserving generalization capability primarily pertains to zero-shot approaches, which was our main focus.
> We acknowledge the relevance of AIM and ST-Adapter in employing a similar learning approach but would like to clarify that these methods are fully supervised, whereas our emphasis is on zero-shot techniques. However, we value your feedback, and in the revised version, we have updated our statement to ensure accuracy and clarity.
> If you have any further questions or suggestions, please feel free to let us know.
>
> **W3.2: Spatial and Temporal Adapter.**
>
> **Answer:**
> Thank you for your thoughtful comments and suggestions. Your observation regarding the comparison of the baseline models is valid, and we would like to provide further insights into our choices.
>
> **1)** AIM as the reviewer rightly pointed out, is a fully supervised approach that excels in supervised classification. In our preliminary experiments, we conducted experiments including both spatial and temporal adapters in the baseline model. However, we observed that this approach tends to overfit to the source dataset, resulting in poor performance on the target dataset, particularly in zero-shot scenarios.
>
>
>
> |               Method                              |HMDB|UCF|K-600|
> |-----------------------------------------------|---------|-------|--------|
> | CLIP(temporal adapters)                |46.0    |70.5 | 61.0  |
> | EZ-CLIP(Temporal visual prompt)  |52.9    |79.1 | 70.1  |
>
>
> **2)** Apart from this performance issue, temporal adapters also suffer from scaling challenges. For a more detailed understanding, we have provided the breakdown of trainable parameters for different configurations, highlighting the scalability challenges associated with temporal adapters, especially with larger models like ViT-14.
> | Model                 | Component                    | Trainable Parameters (M) |
> | ----------------------| -------------------------------- | ---------------------------------- |
> |  CLIP ViT-16      | Temporal Adapters         |                   3.5                   |
> |  CLIP ViT-14      | Temporal Adapters         |                  12.6                  |
>
>
> | Model                 | Component                        | Trainable Parameters (M) |
> | ----------------------|------------------------------------ | ---------------------------------- |
> |    CLIP ViT-16    | Temporal Visual Prompts    |                  0.07                 |
> |   CLIP ViT-14     | Temporal Visual Prompts    |                 0.19                  |
>
>
>
> Additionally, we explored the impact of incorporating temporal visual prompts alongside spatial adapters, and the results indicated challenges in scalability, especially with larger models.
>
> We hope this clarifies our rationale and the considerations we took into account during the experimentation. If you have further questions or suggestions, please feel free to let us know. Your feedback is invaluable in enhancing the quality and clarity of our work.

---

> ### Author Response · Authors · 2023-11-22
> **Responce to Reviewer wVRo[2/3]**
>
> **W3.3 Impact of the LLM-generated action class descriptions.**
>
> **Answer:**
>
> ---
>
> Thank you for your insightful observation. We appreciate your careful examination of our results. The role of LLM-generated action class descriptions is indeed crucial in our approach, and we acknowledge their positive impact on action recognition, as demonstrated in Table 5.
>
> While the use of LLM-generated descriptions contributes to the improved performance of EZ-CLIP, it is not the sole factor. Although the use of LLM-generated descriptions in this context is a novel idea, we would like to emphasize that our approach involves a joint training strategy that incorporates both temporal visual prompting and motion loss. The ablation studies in our paper demonstrate that the introduction of temporal visual prompts and motion loss leads to significant performance gains across various datasets. The use of LLM-generated descriptions without temporal visual prompting and motion loss is not as strong as it is with these components.
>
> *Zero-shot Ablation on LLM-generated Action Class Descriptions:*
>
>
> | TVP  |Motion loss | LLM-description   | HMDB-51 | UCF-101 | K-600  |
> |--------|----------------|-------------------------|---------------|--------------|-----------|
> |    ❌ |       ❌        |     ✔️                     |   49.2        |   75.5     |     63.1  |
> |     ✔️|           ✔️    |   ❌                       |  52.0         |   77.8      |    67.2  |
> |    ✔️ |       ✔️        | ✔️                        |    52.9        |   79.1      |   70.1   |
>
>
>                                                        |  HMDB-51                      |  UCF-101                      |    SSv2                            |
> | TVP|Motion loss|LLM-description| K=2 | K=4 | K=8 | K=16 | K=2 | K=4 | K=8 | K=16 | K=2 | K=4 | K=8 | K=16 |
> |-------|---------------|---------------------|-------|-------|--------|--------|-------|--------|-------|--------|-------|--------|-------|---------|
> |  ❌  | ❌            |           ✔️          | 53.7 |57.9 |61.7  | 63.6  |81.2  |85.5  |86.6  |89.8  |5.7    |6.7    |7.9   |  10.8  |
> | ✔️   |✔️             |             ❌        | 57.0 | 59.1| 65.1 | 65.2  |81.8  |86.6  |88.8  |90.2  |6.1    |8.0    |9.3   | 11.9    |
> | ✔️   |✔️             | ✔️                    |57.3  |61.1 |65.4  |67.7   |84.7  |88.3  |91.3  |92.8  |6.8    |8.7    |9.6   |13.2     |
>
>
>
> For instance, when training the model jointly with both temporal visual prompts and motion loss, we observed substantial improvements, particularly in the case of the SSv2 dataset. This indicates that our proposed techniques play a complementary role in enhancing temporal modeling in videos.
>
> Regarding the comparison with ViFi CLIP, we would like to clarify that our goal is to improve efficiency with a focus on zero-shot action recognition. The use of LLM is indeed a novel idea, and we acknowledge its application in other contexts, as seen in ["VISUAL CLASSIFICATION VIA DESCRIPTION FROM LARGE LANGUAGE MODELS"](https://arxiv.org/pdf/2210.07183.pdf). However, even without LLM, our approach achieves comparable performance to existing works while utilizing very few parameters.
>
> We also appreciate your attention to the breakdown of how motion loss and temporal prompts contribute to our results. The attention visualizations in Figure 4 illustrate the shift in focus from appearance to motion in EZ-CLIP, particularly evident in datasets like UCF-101 and SSv2. This shift aligns with our goal of capturing complex relative motion features, especially beneficial in scenarios where motion cues are critical, as observed in the SSv2 dataset.
>
> We hope this clarifies the contributions of our proposed approach, and we are open to further discussions or clarifications.
>
> Thank you for your valuable feedback.

---

> ### Author Response · Authors · 2023-11-22
> **Responce to Reviewer wVRo[3/3]**
>
> **W.3.4: Frame selection strategy.**
>
> **Answer:**
> Thank you for your observation regarding the frame count difference between EZ-CLIP and other methods. We appreciate your attention to detail.
>
> In Table 6, throughput is calculated based on the number of frames processed per second. Therefore, we think that the efficiency comparison in Table 6 is appropriate, irrespective of whether the model employs 8 frames per video or 32 frames per video. Throughput provides a standardized measure that allows for a fair comparison of efficiency across different methods, accounting for the processing speed regardless of the specific frame count used by each method.
>
> If you have further questions or if there are additional aspects you would like us to address, please feel free to let us know. We value your input and are committed to ensuring clarity in our methodology and results.
>
> **W.3.5: Evaluation of K-600 dataset**
>
> **Answer:**
> Thank you for highlighting the overlap between K-600 and K-400. We agree with the reviewer and acknowledge the importance of addressing this factor in zero-shot learning evaluations.
>
> We use the same setup used by existing works where we performed the zero-shot learning evaluation on K-600 using a disjoint set of classes that are not shared with K-400. This approach ensures a more robust assessment of the models' performance on unseen classes, enhancing the reliability of our findings.
>
> If you have any further suggestions or inquiries, we appreciate your input and are open to addressing any additional concerns you may have.

---

### Official Review · Reviewer_y6Hw · 2023-11-01

**Soundness:** 3 good
**Presentation:** 3 good
**Contribution:** 3 good
**Rating:** 6
**Confidence:** 5

**Summary:**

This paper proposes to adapt CLIP model for video tasks specifically for action recognition in various generalization bench-marking settings. Specifically, two main contributions are proposed to improve CLIP performance on videos. Firstly, temporal visual prompt (TVP) learning is introduced which are learned at the vision encoder. TVP are layer wise prompts that are uniquely added to mean of each frame features and then self-attention mechanism allows the model to learn temporal contexts. Secondly, the authors propose a motion loss, that encourages maximum diversity in terms of variance and pair-wise difference among video frames. This encourages the model to learn distinct embeddings for each frame based on its motion information.

The performance of model is shown across various generalization benchmark settings which it shows improvements over the prior methods with less compute and trainable parameters. Ablations are conducted to show the effect of each component, which provides a broader perspective on the novelty.

**Strengths:**

1) The idea of keeping spatial model frozen and using specific modules and loss functions to improve temporal modeling is encouraging. This allows the community to easily analyze the contributions by the proposed component only. This also results in very light weight adaptation which allows the model to be trained on very less compute resources.

2) The proposed techniques including temporal prompt tuning and motion loss are fairly motivated. Their individual importance has been further validated with proper ablation studies.

3) The proposed framework shows reasonable improvements over the prior methods. Further analysis like tsne plots as well as per-class results shows more accurate effect due to the proposed techniques.

4) Paper is easy to read and well written.

**Weaknesses:**

1) The proposed overall framework seems to be heavily relied on additional training modules and tricks like LLM based prompt ensembling and spatial-language adaptors. However, there is very little detail given on these modules. For example, what kind of adaptors are used, at which place they are incorporated in the model, how many prompts are being used from LLM, illustrations of LLM prompts? It will be better to provide a high level figure diagram which also shows the usage of these additional components.

2) It will be good to see effectiveness of the proposed approach on (i) larger CLIP models like ViT Large or Huge (ii) and on any other CLIP variant (e.g EVA-CLIP) which would confirm the generalization of the proposed approach towards other model scales and recent VL models other than CLIP.

3) I am not completely sure but I think there might be something wrong in the ssv2 results for base-to-novel generalization setting. EZ-CLIP is achieving around 54% and 20.6% accuracy for 16 shots samples. But the same EZ-CLIP is performing relatively poor in few-shot setting where it also uses 16 shot samples. Can the authors revisit the experiments and confirm that they are using the correct split for base-to-novel setting for ssv2? I am afraid it can be the case that complete ssv2 training data is used instead of 16 shots.
Otherwise the few-shot results should have similar scale results for the proposed approach.

**Questions:**

Please refer to the weaknesses section for my queries.

---

> ### Author Response · Authors · 2023-11-22
> **Response to Reviewer y6Hw[1/2]**
>
> We thank the reviewer for their valuable comments and detailed questions. Please refer to our responses below:
>
> **W/Q 2.1: Detail about spatial and text adapters?**
>
> **Answer:**
> We appreciate your inquiry regarding the details of additional training modules and components in our proposed framework. We understand the importance of providing a comprehensive understanding of these elements.
>
> To address your concerns:
>
> 1. **Text and Spatial Adapters:**
>    - The details on the placement of text and spatial adapters can be found in Figure 6, specifically in the Lth block BL, as highlighted in Section A.3 of the revised paper. This illustrates the incorporation of these adapters into our model.
>
> 2. **LLM-Based Prompt Ensembling:**
>    - The usage of prompts from LLM is an integral part of our approach. In Table 7, available in Section A.2, we provide examples of action class descriptions generated by LLM using a template of "describe [category] as an action performed by humans." This demonstrates the effectiveness of LLM-based prompt ensembling in generating diverse and informative descriptions.
>
> We recognize the importance of clarity in visual representation. In response to your suggestion, we will consider incorporating a high-level figure diagram in the revised version of the paper to provide a more intuitive overview of the overall framework, including the usage of these additional components.
>
> Thank you for your understanding, and we welcome any further questions or suggestions.

---

> ### Author Response · Authors · 2023-11-22
> **Response to Reviewer y6Hw[2/2]**
>
> **W/Q 2.2: Experiments on other CLIP models**
>
> **Answer:**
> Thank you for your suggestion. We have indeed conducted experiments on two additional CLIP architectures: CLIP ViT-32 and CLIP ViT-14. These experiments cover three scenarios: (a) Zero Shot, (b) Base to Novel, and (c) Few Shot. Here are the details:
>
> **CLIP ViT-32:**
> - Number of Tunable Parameters:
>   - TVP: 0.07M
>   - Visual Adapters: 3.5M
>   - Text Adapters: 1.5M
>
> **CLIP ViT-14:**
> - Number of Tunable Parameters:
>   - TVP: 0.19M
>   - Visual Adapters: 12.6M
>   - Text Adapters: 3.5M
>
> Zero-Shot experiment
>
> |               Method  | Input size     | HMDB-51  | UCF-101 | K-600 |
> |------------------------|-------------------|---------------|--------------|---------|
> | EZ-CLIP(ViT-32) | 8x224x224   |     50.0       |     77.5     | 67.0  |
> |------------------------|-------------------|---------------|--------------|---------|
> | EZ-CLIP(ViT-16) |  8x224x224  |     52.9       |      79.1    | 70.1  |
> |------------------------|-------------------|---------------|--------------|---------|
> | EZ-CLIP(ViT-16) | 16x224x224 |     53.3       |      80.0    | 71.1  |
> |------------------------|-------------------|---------------|--------------|---------|
> | EZ-CLIP(ViT-14) |  8x224x224  |     55.2       |      82.6    |  72.1 |
>
> Base to Novel experiment
>                               | Kinetics-400            |         HMDB-51       |     UCF-101            |        SSv2               |
> |------------------------|---------|----------|-------|---------|----------|------|---------|----------|------|---------|----------|------|
> |        Method         | Base | Novel  | HM  | Base  | Novel | HM | Base | Novel  | HM | Base | Novel  | HM |
> |------------------------|---------|----------|-------|---------|----------|------|---------|----------|------|---------|----------|------|
> | EZ-CLIP(ViT-32) | 80.3  | 58.16   | 67.4 | 77.0  | 55.3    |64.3 | 95.4  | 70.0    |80.7| 53.1   | 19.1    |28.0|
> |------------------------|---------|----------|-------|---------|----------|------|---------|----------|------|---------|----------|------|
> | EZ-CLIP(ViT-16) | 83.7   | 62.5    | 71.5 | 79.8  | 62.2    |69.9| 95.9   | 76.5    |85.1| 54.0   | 20.6   |29.8 |
> |------------------------|---------|----------|-------|---------|----------|------|---------|----------|------|---------|----------|------|
> | EZ-CLIP(ViT-14) | 85.6   | 67.2    |75.2 | 81.0   | 64.5    |71.8| 96.4   | 79.5    |87.1| 60.5   | 23.0   | 33.3|
>
> Few shot experiment
>
>                              |HMDB-51                       |         UCF-101               |        SSv2                |
> |-----------------------|---------------------------------|--------------------------------|----------------------------|
> | Method                |K=2   | K=4 | K=8|K=16  | K=2 | K=4  | K=8 |K=16| K=2|K=4|K=8  |K=16|
> |------------------------|--------|-------|-------|--------|-------|--------|--------|-------|------|------|------|-------|
> | EZ-CLIP(ViT-32) | 52.4 | 56.9 | 60.4 | 63.7 | 79.7 | 83.7 | 87.5 | 89.8 | 5.8 | 6.7 | 8.0 | 12.1 |
> |------------------------|--------|-------|-------|--------|-------|--------|--------|-------|------|------|------|-------|
> | EZ-CLIP(ViT-16) | 57.3 | 61.1 | 65.4 | 67.7 | 84.7 | 88.3 | 91.3 | 92.8 | 6.8 | 8.7 | 9.6 | 13.2 |
> |------------------------|--------|-------|-------|--------|-------|--------|--------|-------|------|------|------|-------|
> | EZ-CLIP(ViT-14) | 62.4 | 65.3 | 68.4 | 70.7 | 85.2| 89.7  | 92.4 | 93.3 | 8.9 |10.1| 11.1|15.2 |
>
> These experiments provide valuable insights into the effectiveness of our proposed approach across various CLIP model scales. If you have any further inquiries or suggestions, please feel free to let us know.
>
> **W2.3: Few-shot vs base-to-novel comparison on SSv2.**
>
> **Answer:**
> Thank you for your careful examination of our results. To clarify, we employed the same protocol as in existing works for the base-to-novel setting. Specifically, during base-to-novel training, we use all samples from the base classes (not just 16), creating a split between base and novel categories. In the few-shot learning setting, a fixed number of samples (e.g., 16-shot) is utilized.
>
> The observed differences in accuracy between the base-to-novel and few-shot settings, even with the same number of samples (16 shots), are expected due to the distinct training strategies and the resulting differences in the number of training classes between these two setups. We appreciate your attention to detail and hope this explanation clarifies the discrepancy you noticed.
>
> If you have any further questions or suggestions, please feel free to let us know.

---

### Official Review · Reviewer_PcJL · 2023-11-01

**Soundness:** 2 fair
**Presentation:** 3 good
**Contribution:** 2 fair
**Rating:** 5
**Confidence:** 5

**Summary:**

This paper addresses the issues of adapting image-based visual-language models to the video domain and proposes solutions. It introduces a prompt-based temporal modeling method to reduce computational complexity when modeling temporal relationships. Additionally, it designs a motion loss to capture the intrinsic relationships between frames, which existing methods often overlook.

**Strengths:**

+ The paper proposes a prompt-based approach for establishing temporal relationships between different frames in image models.

+ Existing contrastive losses neglect the intrinsic properties of videos. The motion loss is designed to enable the model to learn motion information between frames.

+ The method is simple but has been validated on multiple settings and datasets, achieving state-of-the-art (SOTA) performance.

+ The proposed method outperforms other methods in terms of GFLOPs, throughput, tunable parameters, and total parameters.

**Weaknesses:**

- The contribution of the paper is limited due to similarities with the Visual Prompt Multi-Modal Tracking [A] approach. Both methods utilize prompts and transformer outputs to generate new prompts interactively. The difference lies in this paper's method of averaging patches from each frame and adding them to the prompt, incorporating information from the current frame. From this perspective, the innovation appears somewhat lacking.

- The paper lacks sufficient ablation experiments, as it only conducts them in the "Base to novel generalization" setting.

- Based on the ablation experiments in Table 4 and Table 5, combined with the results in Table 2, it can be observed that even without the proposed TVP and Motion loss, adding only the adapter achieves better results compared to existing methods. Hence, there are concerns regarding the fairness of the comparison with SOTA methods.

[A]  Visual Prompt Multi-Modal Tracking. CVPR23

**Questions:**

- The ablation experiments of LLM were only conducted in the "Base to novel generalization" setting. Were LLM used in other settings? If so, what were the results when LLM was not used?
- The paper mentions the use of text adapter and spatial adapter, citing different papers. However, it is not clear where they are incorporated in Figure 2. Could you clarify where they are added?
- In Table 6, the proposed method has a Tunable params value of only 5.2M. However, the cited Aim's spatial adapter has 3.7M tunable parameters, and LoRA has a minimum of 1.8M. Could you provide the specific breakdown of the tunable parameters for each component?
- For the prompt in this paper, it is added to each layer of the transformer. Has there been an experiment where the prompt is only added to the first layer?
- Motion loss: In Equation (5), the central difference C is computed using the embeddings of the previous and next frames. However, in action recognition tasks, adjacent frames are often very similar. Is this loss calculation effective? Could you explain in detail the role of motion loss?

====================After rebuttal=================
My main concerns about the technical contribution have not been addressed (see my detailed comments). I keep my initial rating.

---

> ### Author Response · Authors · 2023-11-22
> **Response to Reviewer PcJ L[1/4]**
>
> We thank the reviewer for their time, insightful comments, and questions. We have provided our responses below.
>
> **W.1.1 Comparison with Visual Prompt Multi-Modal Tracking [A]**
>
> **Answer:**
> Thank you for sharing this work with us. It is indeed relevant, and we have added this to our related work discussion in the revised version.
>
> We believe ViPT [A] and our proposed temporal visual prompting are significantly different and cannot be directly compared.
>
> **Conceptual Difference (Temporal communication and feature integration):**
> Our temporal visual prompting adapts via communicating across several video frames with the help of prompts, enabling the efficient learning of motion aspects within the video. It is not just combining spatial information but learning temporal dependencies via prompting. This unique aspect of our approach allows us to capture and utilize information from the current frame, significantly enhancing our model's ability to understand the temporal dynamics within videos.
>
> ViPT, on the other hand, is focused on integrating multiple modalities with an MCP module which combines features (addition to be exact, Figure 3 from ViPT) from two different modalities. There is no communication across modalities via prompting; it is rather an aggregation/fusion of features (due to prompts after fusion, not during fusion). We believe that the introduction of TVP represents a substantial contribution to the field, addressing the need for more effective methods in handling temporal aspects in video understanding.
>
> **Practical Difference:**
> Even if we want to use ViPT for videos, it cannot be used for more than 2 frames without any significant modifications to the MCP module. Also, even after this modification, it will be closer to spatial prompting with pooling/fusion, as there is no way to communicate across frames, which is our main technical novelty.

---

> > ### Comment · Reviewer_PcJL · 2023-12-01
> > **Thank you for the response. There are still some concerns.**
> >
> > I have carefully read the authors' responses and the comments from other reviewers.
> >
> > 1. I appreciate the additional experiments on zero-shot and few-shot settings. However, what I would like to see more is some design-based ablation studies regarding TVP. Since the design of TVP shares many similarities with existing methods, it necessitates further experiments to demonstrate that this design is superior to current prompts. It should be shown that some functionalities of this design are unachievable by existing prompts.
> >
> > 2. Although the authors explained how their baseline differs from the compared methods, they did not conduct a fair comparison under the same setting.
> >
> > 3. After reading the authors' response, I fail to discern the fundamental difference between this paper's prompt and ViPT. It seems that the temporal relation between frames can be modeled after passing through the self-attention layer, and it is not a capability unique to having a prompt.

---

> ### Author Response · Authors · 2023-11-22
> **Response to Reviewer PcJ L[2/4]**
>
> **W.1.2 Additional ablation experiments.**
> **Answer:**
> Thank you for your valuable feedback regarding the need for additional ablation experiments. In response to your suggestion, we have conducted comprehensive ablation experiments and have included the results in the revised manuscript. We present the impact of LLM-generated action class descriptions as prompts in various settings, including zero-shot, few-shot, and few-shot ablation with a focus on TVP and motion loss.
> The tables below provide a comprehensive overview of our ablation experiments:
> **1) Impact of LLM-generated action class descriptions as prompts on a Zero-shot setting:**
>
> | LLM-description | HMDB-51 | UCF-101 | K-600  |
> |-----------------------|---------------|--------------|----------|
> |  ❌                     |  52.0         |   77.8       |  67.2   |
> |------------- ---------|---------------|--------------|----------|
> | ✔️                      |  52.9         |  79.1        | 70.1   |
> **2) Zero-shot ablation on TVP and Motion loss:**
> We evaluate the effectiveness of temporal visual prompting (TVP) and motion loss using a Zero-shot setup trained on Kinetics-400 and tested on UCF-101, HMDB-51, and Kinetics-600.
>
> | TVP                     | Motion loss | HMDB-51 | UCF-101 | K-600  |
> |------------------------|-----------------|---------------|--------------|----------|
> | ❌                       | ❌                |  49.2         | 75.5        |  63.1   |
> |------------------------|-----------------|---------------|--------------|----------|
> | ❌                       | ✔️                |  50.0         |75.9         | 65.5    |
> |------------------------|-----------------|---------------|--------------|----------|
> | ✔️                       | ❌                |  51.2         | 78.1        | 67.5    |
> |------------------------|-----------------|---------------|--------------|----------|
> | At first layer only | ✔️                |  50.4         |  76.3       |  65.6   |
> |------------------------|-----------------|---------------|--------------|----------|
> | ✔️                       | ✔️                |  52.9         |  79.1       |  70.1   |
> |------------------------|-----------------|---------------|--------------|----------|
>
> **3) Few-shot ablation on TVP and Motion loss:**
> We evaluate the effectiveness of temporal visual prompting (TVP) and motion loss while using a Few-shot setup on UCF-101, HMDB-51, and SSv2.
>
>                                  | HMDB-51                   | UCF-101                    |    SSv2                             |
> |---------------------------|------------------------------|-----------------------------|-----------------------------------|
> | TVP | Motion loss   |K=2 | K=4  |K=8 |K=16| K=2 |K=4| K=8|K=16  |K=2  | K=4  | K=8 |  K=16  |
> |--------|------------------|------|--------|-------|-------|-------|------|-------|-------|-------|--------|-------|-----------|
> |❌     |         ❌        |53.7|57.9   |61.7  |63.6 |81.2 |85.5|86.6  | 89.8| 5.7   |6.7    |7.9   |   10.8    |
> |--------|------------------|------|--------|-------|-------|-------|------|-------|-------|-------|--------|-------|-----------|
> |❌     |          ✔️       |56.0| 58.5  |64.7  |65.7 |82.1 |84.6|88.5  |90.3 |5.9    |7.7    | 8.7   | 12.0    |
> |--------|------------------|------|--------|-------|-------|-------|------|-------|-------|-------|--------|-------|-----------|
> |✔️     |           ❌      |56.7|60.3   |63.5 |66.7  |83.3 |86.1|89.9  |91.6 |6.6   | 8.1    |9.0   |   12.7    |
> |--------|------------------|------|--------|-------|-------|-------|------|-------|-------|-------|--------|-------|-----------|
> | ✔️    |           ✔️      |57.3|61.1   |65.4 |67.7  |84.7 |88.3|91.3  |92.8 | 6.8   |8.7    | 9.6   |   13.2   |
>
> **Few-shot Ablation LLM:**
> Impact of LLM-generated action class descriptions as prompts on Few-shot settings.
>
>                              |  HMDB-51                       |      UCF-101                   |           SSv2                   |
> |-----------------------|----------------------------------|---------------------------------|---------------------------------|
> |LLM-description  | K=2  | K=4 | K=8 | K=16 | K=2 | K=4 | K=8  | K=16 | K=2 | K=4  | K=8 | K=16 |
> |-----------------------|--------|-------|-------|---------|-------|-------|--------|---------|-------|--------|-------|---------|
> |      ❌                 |57.0   |59.1 | 65.1 |65.2   |81.8 |86.6  |88.8   | 90.2  | 6.1   | 8.0-   |9.3   |11.9   |
> |-----------------------|--------|-------|-------|---------|-------|-------|--------|---------|-------|---------|------|------- -|
> |      ✔️                 |57.3   |61.1 |65.4 |67.7    |84.7  |88.3 |91.3   |92.8    |6.8   | 8.7    | 9.6  |13.2   |
>
>
>
> We hope that these additional ablation experiments address the concern.

---

> ### Author Response · Authors · 2023-11-22
> **Response to Reviewer PcJ L[3/4]**
>
> **W.1.3 Ablation experiments in Table 4 and Table 5.**
>
> **Answer:**
> Thank you for your insightful observation regarding the ablation experiments and the observed performance of our base model in comparison to existing methods.
>
> The base model consists of a spatial adapter for the visual encoder and text encoder with text adapters, where only these adapters are trained, keeping all the CLIP weights frozen. We acknowledge that previous state-of-the-art (SOTA) methods often did not leverage the full potential of pre-trained weights and instead opted for fully training their models. This is a significant difference between our approach and existing methods, and therefore, for some cases, our base model outperforms their performance. This is also evident from fully supervised approaches as well (such as AIM), where freezing the CLIP weights has shown benefits. Even with this strong base model, we do demonstrate the effectiveness of both Temporal Visual Prompting and Motion loss on top of this with a comprehensive set of ablations.
>
> We appreciate your diligence in scrutinizing the experimental setup, and we believe that this clarification provides a more comprehensive understanding of the choices made in our methodology.
>
> **Q1.1 Ablation on LLM in zero shot, base-to-novel, and few-shot.**
>
> **Answer:**
> Thank you for your thoughtful inquiry. We value your meticulous examination of our work and the chance to elaborate on the application of LLM in various settings. Here are the outcomes of ablation experiments conducted using LLM in both zero-shot and few-shot scenarios:
>
> **Zero-shot Ablation on LLM-generated Action Class Descriptions:**
> | LLM-description | HMDB-51 | UCF-101 | K-600  |
> |-----------------|---------|---------|--------|
> |  ❌              |  52.0   |   77.8  |  67.2  |
> | ✔️              |  52.9   |   79.1  |  70.1  |
>
> **Few-shot Ablation on LLM-generated Action Class Descriptions:**
> |-----------------------| HMDB-51                         |        UCF-101                   |          K-600                       |
> | LLM-description | K=2  | K=4  | K=8  | K=16 | K=2  | K=4  | K=8  | K=16 | K=2  | K=4  | K=8  | K=16 |
> |-----------------------|--------|--------|--------|--------|--------|--------|--------|---------|--------|--------|--------|---------|
> |  ❌                     |57.0   |59.1  |65.1   |65.2  |81.8   |86.6   |88.8  |90.2    |6.1     |8.0    |9.3     |11.9    |
> |-----------------------|--------|--------|--------|--------|--------|--------|--------|---------|--------|--------|--------|---------|
> | ✔️                      |57.3   |61.1  |65.4   |67.7  |84.7   |88.3   |91.3  |92.8     |6.8    |8.7    |9.6     |13.2    |
> |-----------------------|--------|--------|--------|--------|--------|--------|--------|---------|--------|--------|--------|---------|
>
> These additional ablation experiments further demonstrate the impact of LLM-generated action class descriptions as prompts in both zero-shot and few-shot scenarios. We hope this clarification addresses your concern, and we welcome any further questions or suggestions.

---

> ### Author Response · Authors · 2023-11-22
> **Response to Reviewer PcJ L[4/4]**
>
> **Q.1.2 Detail of spatial and text adapters and placement.**
>
> **Answer:**
> Thank you for bringing this to our attention. We appreciate your inquiry and apologize for any confusion. Since the incorporation of the text adapter and spatial adapter was not the primary focus of our contribution, this detail was provided in the supplementary (Section A.3).
>
> However, to address this concern, we have updated the figure in the revised version of the paper. The updated details on the placement of the adapters are included in the Lth layer in block BL. The transformer block structure BL in Figure 6 illustrates the integration of the Adapter after the self-attention layer, adapting the CLIP architecture.
>
> We hope that these updates provide more clarity into our approach.
>
> **Q.1.3 Tunable parameter clarification.**
>
> **Answer:**
> We appreciate your attention to detail and the chance to clarify the tunable parameters in Table 6. There seems to be a misunderstanding regarding the cited AIM's spatial adapter, as it does not have 3.7M tunable parameters. The accurate figures are 14.3M for SSv2 and 11M for K-400.
>
> It's crucial to note that AIM's method focuses on video classification in a fully supervised manner, while our proposed method addresses the more challenging task of zero-shot classification. For a detailed breakdown of our tunable parameters:
>
> - Temporal Visual prompts: 73K
> - Text adapters: 1.5M
> - Image spatial adapters: 3.5M
>
> These individual components collectively contribute to the overall efficiency and effectiveness of our model in achieving superior zero-shot action recognition. We hope this clarification addresses your concerns, and we remain open to any further questions or feedback.
>
> **Q.1.4 Prompt only added to the first layer.**
>
> **Answer:**
> Thank you for your thoughtful question. We did conduct experiments with a shallow Temporal Visual Prompt (TVP) in the context of zero-shot ablation. Specifically, in this experiment, TVP was applied only to the first layer of the CLIP ViT transformer.
>
> The results of the zero-shot ablation on TVP and Motion loss are presented in the table below:
>
> | TVP                       | HMDB-51    | UCF-101  | K-600      |
> |--------------------------|-----------------|---------------|--------------|
> | At first layer only    |    50.4         |  76.3         |  65.6        |
> |--------------------------|-----------------|---------------|--------------|
> |  At all layers           |     52.9        |   79.1        |   70.1       |
> |--------------------------|-----------------|---------------|--------------|
>
> These results demonstrate the impact of applying TVP exclusively to the first layer of the transformer, where we observe some drop in performance. We hope this clarification addresses your concerns, and we remain open to any further questions or feedback.
>
>
> **Q.1.5 Motion loss explanation.**
>
> **Answer:**
>
> We appreciate the opportunity to provide clarity on the role of motion loss and address your concerns.
>
> In Equation (5), where the central difference C is computed using the embeddings of the previous and next frames, it's important to note that we deliberately avoid using consecutive frames for motion loss calculation. Instead, we select 8 frames with equal intervals from the video. This intentional choice is made to mitigate the common issue in action recognition tasks where adjacent frames are often very similar. This strategy allows us to capture meaningful temporal dynamics while avoiding redundancy caused by the similarity of consecutive frames. In essence, this also allows us to capture a broader context by sampling frames with some skip rate, enhancing the effectiveness of our motion loss calculation.
>
> Please note that this approach aligns with strategies employed by other state-of-the-art methods in the field for frame sampling from videos.
>
> We appreciate your thoughtful review and the opportunity to provide clarifications. If you have any further suggestions or inquiries, we appreciate your input and are open to addressing any additional concerns you may have.

---

> ### Comment · Reviewer_PcJL · 2023-12-02
> **Thanks for the response. There are still some concerns.**
>
> I have carefully read the authors' responses and the comments from other reviewers.
>
> 1. I appreciate the additional experiments on zero-shot and few-shot settings. However, what I would like to see more is some design-based ablation studies regarding TVP. Since the design of TVP shares many similarities with existing methods, it necessitates further experiments to demonstrate that this design is superior to current prompts. It should be shown that some functionalities of this design are unachievable by existing prompts.
>
> 2. Although the authors explained how their baseline differs from the compared methods, they did not conduct a fair comparison under the same setting.
>
> 3. After reading the authors' response, I fail to discern the fundamental difference between this paper's prompt and ViPT. It seems that the temporal relation between frames can be modeled after passing through the self-attention layer, and it is not a capability unique to having a prompt.

---

### Author Response · Authors · 2023-11-22
**Response to Reviewers**

We express our sincere gratitude to the reviewers for dedicating their valuable time to assess our work and for providing insightful comments.

We are grateful to the reviewers for highlighting the ***simplicity*** (PcJL, wVRo), ***light-weight adaptation*** (PcJL, y6Hw), excellent performance (PcJL, y6Hw, wVRo), and ***comprehensive evaluation*** (PcJL, y6Hw, wVRo) of our proposed method; this is indeed encouraging. We take these comments as a testament to the rigor and significance of our work.
The reviewers have positively identified the ***novel contributions*** of our work, namely, the introduction of two innovative ideas: 1) ***Temporal visual prompting*** and 2) ***Motion-loss.*** We appreciate the recognition of the ***thorough ablations*** performed, encompassing various datasets and tasks with ***state-of-the-art performance*** across all studied tasks and datasets. The reviewers' recognition of the ***efficiency of our solution*** aligns with our intentions to provide a practical and effective approach for zero-shot action recognition.

We have sincerely considered all the concerns raised by the reviewers, and in the revised version we have ***positively addressed*** all those issues to the best of our capacity. We have provided a detailed response to each comment and we request our reviewers to please go through individual responses and let us know if there are any further questions.

Here is a summary (not an exhaustive list) of how we have addressed major concerns raised by reviewers,
- Clarification on differences with referred multi-modal tracking work
- More ablations on zero-shot and few-shot demonstrating effectiveness of temporal visual prompting and motion loss.
- Experiments with other CLIP models to show generalization
- Further analyzing impact of LLM
- More experiments for variations on adding prompts, and both spatial and temporal adapters
- Provided details of text and spatial adapter

Once again, we express our appreciation for the time and effort invested by the reviewers in assessing our submission.
Thank you for considering our responses and recognizing the merits of our contributions.

---

### Meta-Review · Area_Chair_tmMx · 2023-12-06

**Metareview:**

Visual prompting was recently popularized as a low-overhead method to adapt to downstream tasks by introducing (in some way) additional prompts/tokens which can be learned, whereas the rest of the model is kept frozen. The approach was shown to be useful in NLP, and lately in vision on tasks such as video temporal grounding and tracking. The authors transfer some of these ideas to action recognition and introduce an additional "motion loss" to help distill video-specific information into the visual prompts. This results in a model that can be applied to zero-shot and few-shot action recognition.

This is a borderline submission. The authors study an important problem and demonstrate the utility of the proposed approach on some benchmarks. At the same time, the reviewers pointed out several issues with the current manuscript: (1) Novelty with respect to similar work in NLP and vision (tracking, video temporal grounding), and (2) empirical evaluation.

In the empirical evaluation multiple issues were raised (unfair comparison to existing methods, not enough clarity on the exact setup used, and missing comparisons to newer models. While the authors provided some evidence to address (1) and (2), the reviewers were unconvinced.

After reading the manuscript, I maintain that the work needs a major revision to incorporate the new results and adjust some of the (strong) claims brought forward by the authors, with a specific focus on the empirical section.

**Justification For Why Not Higher Score:**

Major issues with the empirical evaluation

**Justification For Why Not Lower Score:**

N/A

---

### Decision · Program_Chairs · 2024-01-16

Reject